# Learning Generalizable Part-based Feature Representation for 3D Point Clouds

**Xin Wei, Xiang Gu, Jian Sun** (✉)
School of Mathematics and Statistics, Xi'an Jiaotong University, P. R. China
`{wxmath, xianggu}@stu.xjtu.edu.cn, jiansun@xjtu.edu.cn`

## Abstract

Deep networks on 3D point clouds have achieved remarkable success in 3D classification, while they are vulnerable to geometry variations caused by inconsistent data acquisition procedures. This results in a challenging 3D domain generalization (3DDG) problem, that is to generalize a model trained on source domain to an unseen target domain. Based on the observation that local geometric structures are more generalizable than the whole shape, we propose to reduce the geometry shift by a generalizable part-based feature representation and design a novel part-based domain generalization network (PDG) for 3D point cloud classification. Specifically, we build a part-template feature space shared by source and target domains. Shapes from distinct domains are first organized to part-level features and then represented by part-template features. The transformed part-level features, dubbed aligned part-based representations, are then aggregated by a part-based feature aggregation module. To improve the robustness of the part-based representations, we further propose a contrastive learning framework upon part-based shape representation. Experiments and ablation studies on 3DDA and 3DDG benchmarks justify the efficacy of the proposed approach for domain generalization, compared with the previous state-of-the-art methods. Our code will be available on http://github.com/weixmath/PDG.

## 1 Introduction

The 3D shape understanding and reasoning play a critical role in wide applications such as automatic drive, archaeology, virtual reality / augmented reality, *etc*. Point cloud is a popular representation of 3D shape attributed to its simpleness and effectiveness. Recently, with the thriving of deep learning, numerous deep architectures [1–10] have been proposed for 3D point cloud analysis. Although these methods have achieved impressive results on the 3D shape classification task, all of them strongly rely on the i.i.d assumption on the source and target data but ignore the out-of-distribution situation in real-world practice. For example, the performance of these methods drops dramatically when trained on CAD datasets [11, 12] and tested on real scanned datasets [13, 14]. Therefore, it is crucial and valuable to investigate how to learn a 3D classification model that generalizes well on a related test domain with domain shift to the training domain [15]. Domain shift refers to the existence of significant divergence between the distributions of the training and test datasets [15]. The domain shift may degrade the performance of network trained on a training dataset when generalizing to the test dataset. Domain generalization methods for improving domain robustness of deep network have been extensively investigated for 2D images [16–21]. However, little work [22, 23] addresses the domain generalization (DG) problem for point cloud deep networks. For point clouds, the major cause of domain shift is the geometry variations generated by inconsistent data collection processes, *e.g*., the realistic sensor noises, the non-uniform density of point clouds, and self-occlusion. These domain shifts specific for 3D point clouds hinder the idea that directly adopts 2D image-oriented

DG methods to point clouds, and this inspires us to design a 3D domain generalization model for addressing the domain shifts of point clouds.

To improve the model robustness to geometry variations, a straightforward way is to simulate geometry shift during training. MetaSets [22] designs three point transformation tasks of simulating occlusions, missing parts and changes in scanning density, which have been proven to be effective as data augmentations. They further learn feature representations by meta-learning using these tasks. Instead of improving model's generalization ability using globally pooled point cloud features, in this paper, we tackle the domain generalization problem for 3D point clouds based on our observation that the local geometric structures of point clouds are more likely to be shared across distinct domains, therefore being more generalizable to the geometry variations.

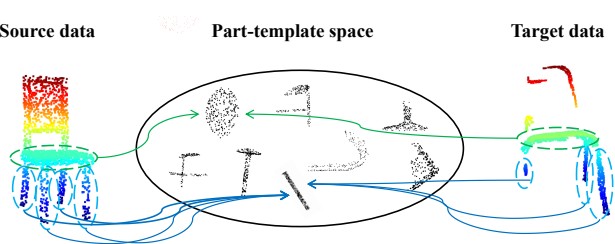

Figure 1: The 3D point clouds in source domain (*i.e.*, source data on the left) and target domain (*i.e.*, target data having missing points on the right) can be both recognized as "chair" because they can be represented by the shared parts in the part-template space of "chair" category.

Parsing objects into parts is crucial for humans to understand the world. For the task of object recognition, the visual system of people decomposes shapes into parts and recognizes objects based on the description and spatial relations of parts [24]. It is hard for machines to understand the chair on the right of Fig. 1 when it overfits on a set of chairs similar to the left one in Fig. 1. However, humans can easily interpret them by decomposing two chairs and inferring their labels from some similar parts (*e.g.*, legs and a plain), even when having broken parts of the right chair in Fig. 1. This motivation seems to be intuitive, and we will provide more experimental evidence in Sect. 2. Based on this observation, we are interested in learning generalizable representation of 3D point clouds at the part-level instead of the global shape-level.

Along this idea, in this work, we propose a novel part-based domain generalization network for 3D point cloud classification. We build a part-template feature space that is shared to source and target domains. Shapes from distinct domains are first organized as part-level features and then aligned to part-template features by a cross attention mechanism. Aligned part-based features are then aggregated by a part-based feature aggregation module for each point cloud. To improve the robustness of part-based representation, we further propose a contrastive learning framework to enforce that the feature representations of a point cloud under different transformations are consistent in part-level and shape-level. Extensive experiments conducted on 3DDA [25] and 3DDG [22] benchmarks demonstrate the effectiveness of our approach, and our method outperforms the compared methods by a notable margin.

Our contribution can be summarized in three folds. *First*, we empirically observe that the geometry shift induced domain gap of point clouds could be reduced by part-level representation, thus we propose to learn part-based 3D feature representation to improve the generalization ability of point cloud classification models. *Second*, we propose a novel part-based domain generalization network for 3D point cloud classification. A contrastive learning framework upon part-based shape representation is further designed to improve the robustness of learned representations. *Third*, our method achieves the best performance by comparisons on 3D benchmarks for domain generalization.

## 2   Reducing Geometry Shift by Part-based Feature Representation

**Problem definition.**   Let $\mathcal{X}$ and $\mathcal{Y}$ be input and label spaces. A domain is defined by $(\mathcal{D}, g_{\mathcal{D}})$ where $\mathcal{D}$ is a probability distribution on $\mathcal{X}$ and $g_{\mathcal{D}} : \mathcal{X} \to \mathcal{Y}$ is a function mapping input to its ground-truth label. The objective of domain generalization is to train a classifier $\mathcal{F}$ on the source domain $\mathcal{D}^S$ that predicts well on target domain $\mathcal{D}^T$ when the target samples are not available during training. In general, we assume the label space is shared by source and target domains.

In 3D domain generalization, the input is a point cloud $P \in R^{N \times 3}$, where $N$ is the number of points. We consider a point cloud classification model $\mathcal{F}$ composed of a feature extractor $f_\theta$ :

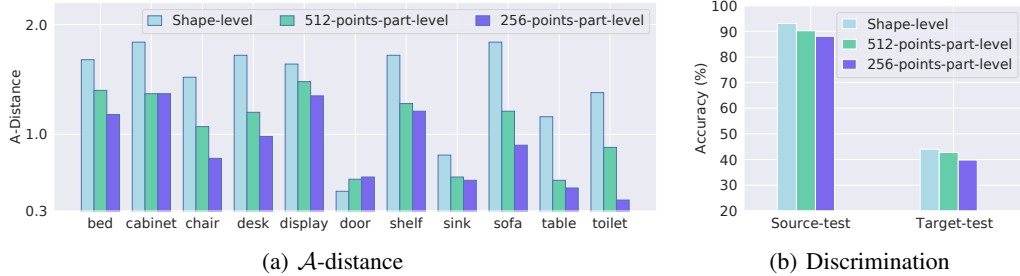

|(a) $\mathcal{A}$-distance | (b) Discrimination |

Figure 2: (a) $\mathcal{A}$-distance between source and target domain features (shape-level, 512-points-part-level, 256-points-part-level) for each class. (b) Discrimination ability of shape-level features and part-level features measured by classification accuracy on source and target domains.

$P \rightarrow Z$ and a classifier $C_\psi : \mathbf{F} \rightarrow R^C$, where $Z \in R^{N \times d}$ is point-wise features, $\mathbf{F} \in R^d$ is a global representation of $P$, and $C$ is the number of classes. The final prediction is given by $\hat{y} = \text{softmax}(C_\psi(\text{pooling}(f_\theta(P))))$, where the pooling operation is determined by $\mathcal{F}$. The parameters $(\theta, \psi)$ are optimized with respect to the cross entropy loss $\mathcal{L}(y, \hat{y}) = -\sum_{i=1}^{C} y_i \cdot \log(\hat{y}_i)$. Directly minimizing the cross entropy loss $\mathcal{L}$ will produce a discriminative network in the source domain, however, it may overfit the source domain and degeneralize on the target domain.

**Comparison of generalization and discrimination abilities of shape-level and part-level features.**
To understand a shape, people may process it in various granularities. The fine-grained parts represent local geometric structures while coarsen parts contain more global semantic information. Thus global shape-level features may be more discriminative than part-level features. However, this advantage drops sharply when encountering a large distribution discrepancy between training and test domains for shape-level features. On the contrary, part-level features encode the local geometric structures which are shared across different shapes in distinct distributions, while they are short of semantic information. We next experimentally verify the above analysis. We use $\mathcal{A}$-distance as a measure to evaluate distribution discrepancy [26, 27]. It is defined as $\text{dist}_\mathcal{A} = 2(1 - 2\epsilon)$, where $\epsilon$ is the test error of a classifier trained to discriminate the source and the target domain data. We train a PointNet [5] on source domains $M$ (ModelNet dataset [12]) and test on target domain $SO$ (ScanObjectNN [14]). We first get point-wise features $Z \in R^{N \times d}$ for each shape that has $N$ points. Then point-wise features are max-pooled on all points to produce a global shape-level feature $Z^s$. We also split each shape into some overlapped parts in distinct scales, $i.e.$, 16 parts (256 points in each part) and 8 parts (512 points in each part) and max-pooled on each part to get part-level features $\{Z_i^{sp}\}_{i=1}^{16}$ and $\{Z_i^{lp}\}_{i=1}^{8}$. Fig. 2 (a) shows the $\mathcal{A}$-distance on each class of $M \rightarrow SO$ with shape-level features $Z^s$ and part-level features $\{Z_i^{sp}\}_{i=1}^{16}$ and $\{Z_i^{lp}\}_{i=1}^{8}$. We observe that $\text{dist}_\mathcal{A}$ of part-level features is smaller than $\text{dist}_\mathcal{A}$ of shape-level features and $\text{dist}_\mathcal{A}$ decreases as the size of parts becomes smaller, indicating that part-level features are able to better reduce domain gap. We also train a linear SVM on source domain training data and test on source domain test data and target domain test data to evaluate the discrimination abilities. For part-level features, the prediction of a shape is by voting on the predictions of parts. As shown in Fig. 2 (b), the discrimination abilities of features are weakened when the scale decreases. Though part-level features could reduce the geometry shift between source and target domains, aggregating them simply by voting could not improve the discrimination ability.

Based on the above observations, we are inspired to represent a point cloud by part-level features to reduce the domain discrepancy. We build a part-template feature space that is shared with source and target domains. Part-level features in distinct domains are aligned to the part-template features, resulting in part-based feature representations with better generalization. To improve the discrimination of the part-level features, the features of parts are fused with a part-based feature aggregation module to achieve a global representation for each point cloud.

## 3 Learning Part-based Representation of Point Clouds

We first introduce our proposed part-based representation of point clouds, taken as the main operation of our part-based domain generalization network for 3D point cloud classification presented in Sect. 4.

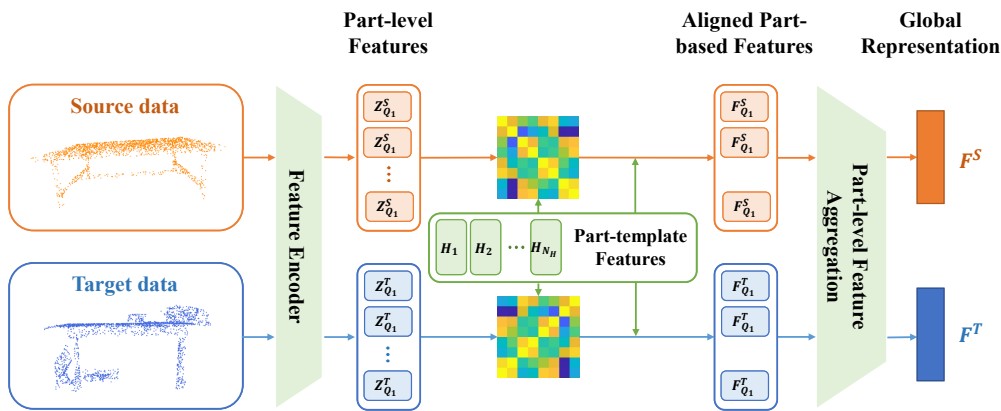

Figure 3: Illustration of our part-based feature representation for point clouds. Given a point cloud $P \in R^{N \times 3}$ from source or target domain, it is first processed by feature encoder and organized to part-level features $Z_Q = \{Z_{Q_i}\}_{i=1}^M \in R^{M \times d}$. Then part-level features are transformed to aligned part-based features $F_Q = \{F_{Q_i}\}_{i=1}^M$ by aligning them to part-template features $H = \{H_i\}_{i=1}^{N_H}$. Then they are aggregated to a global representation $\mathbf{F}$ by the part-based feature aggregation module.

The major objective of this representation is to transform part-level features of shapes in distinct domains into a common space, *i.e.*, part-template feature space. All these part-level features are aligned to a set of learnable part-template features in the part-template feature space. The transformed features are dubbed as the aligned part-based representations of point clouds.

As shown in Fig 3, taking two point clouds from source and target domains, they are first organized to part-level features, and then transformed to aligned part-based representations in part-template feature space by cross-attention. In order to aggregate the aligned part-based representations to a global representation, we further propose a part-based feature aggregation module to aggregate features according to their importance.

## 3.1  From point clouds to part-level features

Given a point cloud $P = \{x_1, x_2, ..., x_N\} \in R^{N \times 3}$, point-wise features $Z = \{z_1, z_2, ..., z_N\} \in R^{N \times d}$ are first extracted by a feature extractor $f_\theta$, *i.e.*, $Z = f_\theta(P; \theta_f)$. We represent a point cloud by a union of $M$ overlapped parts $P = Q_1 \cup Q_2 \cup ... \cup Q_M$, where each part $Q_i = \{x_{i1}, ..., x_{ik}\} \in R^{k \times 3}$ is defined as a center point $x_{i1}$ with its $k$ nearest neighbor points $\{x_{i1}, ..., x_{ik}\}$. We use farthest point sampling (FPS) to sample $M$ center points for constructing $M$ parts since FPS has better coverage of the entire point cloud. For each part $Q_i$, corresponding part-level feature $Z_{Q_i} \in R^d$ is derived by maxpooling on point-wise features of part $Q_i$:

$$Z_{Q_i} = \text{maxpooling}\{z_{i1}, ..., z_{ik}\}. \tag{1}$$

## 3.2  Representing part-level features using part-template features

As discussed in Sect. 2, part-level features reflect the local geometric structures of 3D shapes, which could reduce geometry shifts more effectively. Thus we are inspired to use this general information to improve the generalization ability of the classification model. Given part-level features in the source domain, part-level geometry priors, modeled as part-template features, are extracted from the source domain and provide references for the target domain. Specifically, we could construct a part-template space $H$, where $H = \{H_i\}_{i=1}^{N_H} \in R^{N_H \times d}$ is a set of part-template features. $H$ serves as a basis set and each basis vector could encode a local geometric prior. Our proposed part-template features are similar to the dictionary of local geometric priors and each input part-level feature is represented as a combination of them. A common choice to get a dictionary is to apply k-means clustering on part-level features of the entire training data in each training step. However, the dictionary computed by k-means can only reflect local geometric prior in the current step. Instead, we propose to use a set of learnable part-template features to represent various shapes in the whole training process, which can be updated automatically by gradient propagation. Leveraging this simple yet effective representation, we could capture rich part-level geometry priors.

Given part-level features $Z_Q = \{Z_{Q_i}\}_{i=1}^M \in R^{M \times d}$ and part-template features $H = \{H_i\}_{i=1}^{N_H} \in R^{N_H \times d}$, we want to find a feature transformation $\mathcal{T} : R^{N_H \times d} \to R^{M \times d}$ for representing part-level features with the part-template features, defined as

$$F_{Q_i} \triangleq \mathcal{T}(H)_i = \sum_j \mathbf{T}_{ij} H_j, \forall i = 1, ..., M, \tag{2}$$

where $\mathbf{T} \in R^{M \times N_H}$ is a linear transform map implementing $\mathcal{T}$ and $F_Q \triangleq \{F_{Q_i}\}_{i=1}^M$ is transformed features, dubbed aligned part-based features. Briefly speaking, each aligned part-based feature is a linear combination of part-template features, and the combination weight is decided by the relation between part-level features and part-template features.

Recently, some researchers [28–31] have adopted cross-attention to align features from different modalities. [32, 33] further apply cross attention module to solve the unsupervised domain adaptation problem based on its great power in feature alignment and robustness to noises. In our case, we use cross-attention module to learn dependencies between part-level features and part-template features. The transformation map $\mathbf{T}$ is defined as

$$\mathbf{T} = \text{softmax}(\frac{Z_Q H^T}{\sqrt{d}}), \tag{3}$$

where the softmax function is applied on the rows of the scaled similarity matrix. The $i$-th row of $\mathbf{T}$ is a normalized similarity vector between part-level feature $Z_{Q_i}$ and all the part-template features $\{H_i\}_{i=1}^{N_H}$. Part-template features which are more similar to part-level feature $Z_{Q_i}$ will contribute more to the combination, while the less similar part-template features make little impact.

Optimal transport [34] is another possible approach to distribution transport and alignment. We can use optimal transport to learn $\mathbf{T}$ for transporting the part-template features such that the transported feature distribution is the same as that of part-level feature distribution. We conduct the experiment that PDG with OT for feature alignment in the ablation study. In this work, we employ cross-attention for aligning part-level features to part-template features because of its better performance and lower computation cost.

Once the $\mathbf{T}$ is derived, the corresponding aligned part-based features are derived by $F_Q = \mathbf{T}H$. For each part-level feature $Z_{Q_i}$, we can observe that the aligned part-based feature $F_{Q_i}$ is a weighted sum of part-template features $\{H_i\}_{i=1}^{N_H}$.

### 3.3 Part-level features aggregation

After getting a set of aligned part-based features, a global representation is often obtained by pooling these features. However, various parts of a shape may have different contributions for shape recognition. Given the part-level representation $F_Q = \{F_{Q_i}\}_{i=1}^M$, we define the score of each part as

$$s_i = \max(\mathcal{S}(F_{Q_i}; \theta_{P_s})), \tag{4}$$

for $i = 1, ..., M$. $\mathcal{S}(\cdot)$ is a part score module with parameters $\theta_{P_s}$, outputting the vector of probabilities of a part belonging to $C$ shape classes, and max operation returns the maximal value of this vector. $P_s(\cdot)$ is implemented as the same structure as classifier $C_\psi(\cdot)$. In a word, the score $s_i$ indicates the confidence score for classification of part $F_{Q_i}$. Then each part is weighted by this part score, and the global representation of shape is derived by

$$\mathbf{F} = \text{maxpooling}\{s_1 \cdot F_{Q_1}, ..., s_M \cdot F_{Q_M}\}. \tag{5}$$

## 4 Contrastive Learning Upon Part-based Shape Representation

We build a contrastive learning framework upon part-based shape representation to improve the robustness of the learned aligned part-based features. The main idea is to maximize the agreement between a same shape under different augmentations in part-level and shape-level via contrastive loss. We randomly sample a mini-batch of $N$ point clouds $\{P_k\}_{k=1}^N$ and the contrastive learning task is defined on $2N$ point clouds which are the pairs of samples under different augmentations. We use three point cloud transformations including non-uniform density, dropping, and self-occlusion. For more details, please refer to [22]. For each point cloud $P_k$, we denote $\tilde{P}_k$ as the same point cloud with

another augmentation. Augmented shapes are encoded to derive the aligned part-based representation $F_Q \in R^{2N \times M \times d}$ and global representation $\mathbf{F} \in R^{2N \times d}$. Following [35], they are further processed by a projection head $Proj(\cdot)$, resulting in $\hat{F}_Q$ and $\hat{\mathbf{F}}$. $Proj(\cdot)$ is implemented as a two-layer MLPs with $4d$ hidden dimensions. We adopt the contrastive loss as the form of InfoNCE [36], which learns representation by attracting positive samples and dispelling negative samples. The shape-level contrastive loss is defined as a supervised contrastive loss [37]:

$$\mathcal{L}_{shape}^C = \sum_{k=1}^{2N} \log \frac{\sum_{\hat{\mathbf{F}}_k^+} \exp(sim(\hat{\mathbf{F}}_k \cdot \hat{\mathbf{F}}_k^+)/\tau)}{\sum_{\hat{\mathbf{F}}_k^+} \exp(sim(\hat{\mathbf{F}}_k \cdot \hat{\mathbf{F}}_k^+)/\tau) + \sum_{\hat{\mathbf{F}}_k^-} \exp(sim(\hat{\mathbf{F}}_k \cdot \hat{\mathbf{F}}_k^-)/\tau)}, \quad (6)$$

where $sim(\cdot, \cdot)$ denotes the cosine similarity of two inputs, and $\tau$ is a temperature hyper-parameter, empirically set to 0.07. $\hat{\mathbf{F}}_k$ is the anchor, $\hat{\mathbf{F}}_k^+$ denotes $\hat{\mathbf{F}}_k$'s positively paired sample (excluding anchor itself) with the same label as $\hat{\mathbf{F}}_k$, and the negative sample $\hat{\mathbf{F}}_k^-$ is with different label to $\hat{\mathbf{F}}_k$. By this shape-level contrastive loss, the global representation of same class under different transformations will cluster together while those from different classes will be pushed apart.

Part-level contrastive loss $\mathcal{L}_{part}^C$ is defined in a self-supervised form:

$$\mathcal{L}_{part}^C = \sum_k^{2N} \sum_i^M \log \frac{\exp(sim(\hat{F}_{Q_{k,i}} \cdot \hat{F}_{Q_{k,i}}^+)/\tau)}{\exp(sim(\hat{F}_{Q_{k,i}} \cdot \hat{F}_{Q_{k,i}}^+)/\tau) + \sum_{\hat{F}_{Q_{k,i}}^-} \exp(sim(\hat{F}_{Q_{k,i}} \cdot \hat{F}_{Q_{k,i}}^-)/\tau)}. \quad (7)$$

For the $i$-th part of point cloud $P_k$, whose aligned part-based feature is $\hat{F}_{Q_{k,i}}$, the positive sample $\hat{F}_{Q_{k,i}}^+$ is the aligned part-based feature of $i$-th part of $\tilde{P}_k$. The positive part has the same center point as the anchor part. The negative samples are the other aligned part-based features in this batch. Part-level contrastive loss $\mathcal{L}_{part}^C$ encourages learned aligned part-based feature of a part of shape under different local transformations to be consistent in the feature space.

By the contrastive tasks in both part-level and shape-level, aligned part-based representations are robust to varying local point transformations and the combination of these aligned part-based representations could be discriminative for shape classification.

### 4.1 Overall loss

The overall training loss consists of shape loss $\mathcal{L}_{shape}$, part loss $\mathcal{L}_{part}$, shape-level contrastive loss $\mathcal{L}_{shape}^C$, and part-level contrastive loss $\mathcal{L}_{part}^C$. Given global shape representation $\mathbf{F}$, it is sent to a classifier $C_\psi$ followed by a softmax layer. The total training loss is

$$\mathcal{L} = \mathcal{L}_{shape}(C_\psi(\mathbf{F}), y) + \lambda_p \sum_{i=1}^M \mathcal{L}_{part}(\mathcal{S}(F_{Q_i}), y) + \lambda_C(\mathcal{L}_{shape}^C + \mathcal{L}_{part}^C), \quad (8)$$

where $y$ is the class label and $\mathcal{L}_{shape}$ is a cross entropy loss based on global shape representation $\mathbf{F}$. $\mathcal{L}_{part}$ is the cross-entropy loss defined on aligned part-based feature $\mathcal{S}(F_{Q_i})$. $\mathcal{L}_p$ enforces the part score module $\mathcal{S}(\cdot)$ could discriminate the shape category based on the aligned part-based features.

## 5 Experiments

### 5.1 Datasets

**Sim-to-Real [22].** Real-to-Sim [22] is a 3DDG benchmark consisting of three domains: ModelNet [12], ShapeNet [11] and ScanObjectNN [14]. (a) ModelNet ($M$) contains 12,311 clean 3D CAD models from 40 categories, with 9,483 training models and 2,468 test models. (b) ShapeNet ($S$) contains 51,162 3D models categorized into 55 classes and its objects have larger structure variances compared with ModelNet. (c) ScanObjectNN ($SO$) is a recently proposed real-world 3D object classification dataset with scanned indoor scene data. It contains 2,902 object instances from 15 categories. ScanObjectNN offers more practical challenges including background occurrence, object partiality, and different deformation variants. Two versions of ScanObjectNN, respectively with background (ScanObjectNN-BG, denoted as

$SO_B$) and with hardest perturbations (ScanObjectNN-T50-RS, denotes as $SO_H$), are also utilized for testing. Then six synthetic-to-real point cloud domain generalization tasks are built. $M \rightarrow SO$, $M \rightarrow SO_B$, and $M \rightarrow SO_H$ refer to training network on ModelNet and test on ScanObjectNN, ScanObjectNN-BG, and ScanObjectNN-T50-RS where they have 11 shared classes, while $S \rightarrow SO$, $S \rightarrow SO_B$ and $S \rightarrow SO_H$ are defined as generalizing network from ShapeNet to ScanObjectNN, ScanObjectNN-BG, and ScanObjectNN-T50-RS within 9 shared classes.

We follow the data preparation and experiment setting in [22]. Specifically, we use the official training and test split strategy for each dataset. Each point cloud from three domains contains 2,048 points and is normalized within a unit ball. We report two versions of results, *i.e.*, "Best" that is the best achievable results among the epochs in training a network, while "Last five" indicating the average test results of the last five epochs of the training procedure. Note that the reported "Best" is unreasonable since the labels of test data are inaccessible, but it provides an upper-bound of performance for a domain generalization approach. Thus we mainly report the "Last five" as the comparison measure in this paper. We calculate the average of "Last five" of four DG tasks to evaluate the performance of methods. We conduct each experiment three times and report the average top-1 classification accuracy in tables.

**PointDA [25].** PointDA [25] dataset is a widely used point cloud domain adaptation benchmark, which collects shapes of 10 shared classes from ModelNet [12] (M), ShapeNet [11] (S), and Scan-Net [13] ($S^\star$). Six point cloud domain adaptation tasks including $M \rightarrow S$, $M \rightarrow S^\star$, $S \rightarrow M$, $S \rightarrow S^\star$, $S^\star \rightarrow M$, and $S^\star \rightarrow S$ are built upon PointDA [25]. We follow the data preparation, dataset splitting, experiment setting in [25]. Each point cloud in PointDA [25] has 1,024 points. We implement our method on this benchmark under the setting of domain generalization.

## 5.2 Compared methods

For Sim-to-Real, we first compare our method with three state-of-the-art point classification methods including PointNet [5], DGCNN [6] and PointMLP [4]. We find that these methods achieve better results than the results reported in [22], if a random jittering [5] is performed on the training data which is widely used in 3D classification model training. We further compare our method with state-of-the-art 3DDG method MetaSets [22]. We report their results by directly running their published codes of http://github.com/thuml/Metasets.

For PointDA [25], we compare PDG with current 3D domain adaptation methods which use target domain data in training procedure, including DANN [38], PointDAN [25], RS [39], DefRec + PCM [40], GAST [41] and GLRV [42]. We also compare it with state-of-the-art 3DDG method, *i.e.*, MetaSets [22], in which target domain data is inaccessible during training.

## 5.3 Implementation details

For our PDG, we adopt PointNet and DGCNN as backbones of feature extractor $f_\theta$ and classifier $C_\psi$. For fair comparison, we do not change the architecture of backbone and train all methods except MetaSets for 160 epochs with batch-size 32 on one NVIDIA V100 GPU. MetaSets is trained for 200 epochs with batch-size 32 on two NVIDIA V100 GPUs. We use Adam as the optimizer. The initial learning rate and weight decay are $10^{-3}$, $10^{-4}$. The learning rate reduced to $10^{-5}$ following a cosine quarter-cycle. The part number $M$, points number $k$ in each part, and the number of part-template features $N_H$ are respectively set to 8, 512, 384. $\lambda_p$ and $\lambda_C$ in training loss are 0.05 and 0.01.

## 5.4 Experimental Results

**Comparison with single domain state-of-the-art methods.** As shown in Table 1, PDG with DGCNN as backbone outperforms previous state-of-the-art 3D classification methods by more than 5.5% in average accuracy of four tasks. PointMLP [4] is a powerful method and achieves state-of-the-art results on ModelNet40 (94.1%) and ScanObjectNN (83.9%) for single domain point cloud classification, as shown in their paper. Compared with it, our PDG (DGCNN) shows better generalization ability and outperforms them by 5.5%. Compared with PointNet [5] and DGCNN [6], PDG with PointNet and DGCNN as backbone networks surpass them on every 3DDG task, improving average accuracy by 4.2% and 6.2% respectively. We also find that DGCNN [6]

Table 1: Test accuracy (in %) on Real-to-Sim dataset. We report the "Last five" of each methods as main results for comparison. "Best" of each methods are also presented in bracket.

| Method | $M \to SO$ | $M \to SO_B$ | $M \to SO_H$ | $S \to SO$ | $S \to SO_B$ | $S \to SO_H$ | Avg |
|---|---|---|---|---|---|---|---|
| PointNet [5] | 59.8 (61.5) | 51.5 (53.8) | 50.0 (51.9) | 55.9 (57.4) | 51.0 (54.0) | 49.0 (50.9) | 52.9 (54.9) |
| DGCNN [6] | 60.2 (62.3) | 54.7 (58.5) | 49.2 (51.1) | 54.9 (59.0) | 51.9 (54.3) | 47.2 (54.0) | 53.0 (56.5) |
| PointMLP [4] | 59.0 (62.5) | 58.6 (59.4) | 47.9 (51.1) | 53.7 (57.3) | 55.3 (56.8) | 47.4 (50.9) | 53.7 (56.3) |
| MetaSets (P) [22] | 60.3 (66.3) | 52.4 (57.5) | 47.4 (54.4) | 51.8 (55.3) | 44.3 (50.3) | 45.6 (50.0) | 50.3 (55.6) |
| MetaSets (D) [22] | 58.4 (64.2) | 59.3 (60.6) | 48.3 (53.1) | 49.8 (60.3) | 47.4 (57.8) | 42.7 (50.8) | 51.0 (57.8) |
| **PDG (P)** | **67.6** (**69.4**) | 58.5 (61.1) | **56.6** (57.2) | 57.3 (61.8) | 51.3 (55.5) | **51.3** (53.9) | 57.1 (59.8) |
| **PDG (D)** | 65.3 (68.8) | **65.4** (**68.0**) | 55.2 (**58.0**) | **59.1** (**64.3**) | **59.3** (**64.3**) | 51.0 (**56.6**) | **59.2** (**63.3**) |

Table 2: Classification accuracy (in %) of various 3DDA and 3DDG methods on PointDA-10 dataset. Results of methods which do not use target data during training are in bold.

| Method | DA / DG | $M \to S$ | $M \to S^\star$ | $S \to M$ | $S \to S^\star$ | $S^\star \to M$ | $S^\star \to S$ | Average |
|---|---|---|---|---|---|---|---|---|
| Supervised | - | 93.9 | 78.4 | 96.2 | 78.4 | 96.2 | 93.9 | 89.5 |
| **w/o Adaptation** | - | 83.3 | 43.8 | 75.5 | 42.5 | 63.8 | 64.2 | 62.2 |
| DANN [38] | | 74.8 | 42.1 | 57.5 | 50.9 | 43.7 | 71.6 | 56.8 |
| PointDAN [25] | | 83.9 | 44.8 | 63.3 | 45.7 | 43.6 | 56.4 | 56.3 |
| RS [39] | DA | 79.9 | 46.7 | 75.2 | 51.4 | 71.8 | 71.2 | 66.0 |
| DefRec + PCM [40] | | 81.7 | 51.8 | 78.6 | 54.5 | 73.7 | 71.1 | 68.6 |
| GAST [41] | | 84.8 | 59.8 | **80.8** | 56.7 | **81.1** | 74.9 | **73.0** |
| GLRV [42] | | **85.4** | **60.4** | 78.8 | **57.7** | 77.8 | **76.2** | 72.7 |
| **MetaSets [22]** | DG | **86.0** | 52.3 | 67.3 | 42.1 | 69.8 | **69.5** | 64.5 |
| **PDG (Ours)** | | 85.6 | **57.9** | **73.1** | **50.0** | **70.3** | 66.3 | **67.2** |

outperforms PointNet [5] by only 0.1% while PDG (DGCNN) outperforms PDG (PointNet) by 2.1%. This demonstrates that, given the part-level features better encoding local geometric structures, our aligned part-based representations could further improve the robustness and discrimination ability. These significant performance gains (4.2% and 6.2%) of PDG over the baseline prove the effectiveness of our designed part-based domain generalization network, which improves both the generalization and discrimination abilities of point cloud classification models.

**Comparison with 3DDA and 3DDG methods.** For Sim-to-Real [22], MetaSets [22] is a state-of-the-art 3D domain generalization model which meta-learns point cloud representations from some point transformation tasks. As shown in Table 1, with the same backbone network PointNet and data augmentation, PDG outperforms MetaSets by 6.8%. When equipped with DGCNN, the improvement is 8.2%. In contrast with MetaSets which aims to improve the generalizable ability of shape-level features, our proposed PDG reduces geometry shift more effectively by the part-based feature representation.

For PointDA [25], from Table 2, PDG improves the baseline methods by 5.0% in the average accuracy of all tasks and outperforms 3DDG method Metasets [22] by 2.7%. Specifically, PDG beats MetaSets in four tasks $M \to S^\star$, $S \to M$, $S \to S^\star$, and $S^\star \to M$. For two synthetic-to-real tasks $M \to S^\star$ and $S \to S^\star$, the improvements are 5.6% and 7.9%. We can also find that PDG exceeds some 3DDA methods including DANN [38], PointDAN [25], and RS [39].

## 5.5 Ablation study

In this section, we take a closer look at the effects of components of our network, including cross-attention module for feature alignment, part-level aggregation module and contrastive learning framework upon part-based shape representation. We choose PointNet [5] as backbone network referred as "baseline" and experiments are conducted on $M \to SO$ task on Real-to-Sim [22] dataset. We also investigate the effects of the scale of the parts and visualize the both part-level features and shape-level features of the baseline and our PDG. Results of various architectures of PDG are presented in Table 3.

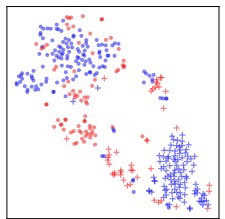 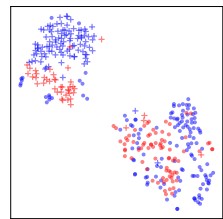 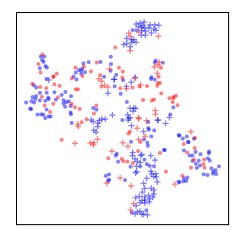 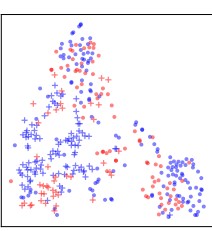

(a) Shape-level (PointNet) (b) Shape-level (PDG) (c) Part-level (PointNet) (d) Part-level (PDG)

Figure 4: (a)(b) The t-SNE [43] visualizations of shape-level features of PointNet and our PDG (PointNet as backbone). (c)(d) The part-level features of PointNet, and aligned part-based features of our PDG (PointNet as backbone). In each figure, the features of "chair" and "sofa" are denoted as "•" and "+" respectively, while data from source ($M$) and target ($S^\star$) are in blue and red.

**Effectiveness of each component.** We first replace cross-attention module with optimal transport to find the transformation map $\mathbf{T}$ for feature alignment as discussed in Sect. 3.2. Table 3 shows the results of PDG employing optimal transport for feature alignment, which is denoted as PDG-OT. Compared with PDG-OT, PDG with cross-attention improves the performance by 2.4%. As mentioned in Sect. 3.3, we design a part-based feature aggregation module to aggregate the aligned part-based features according to their confidence score for classification. We compare it with two baseline pooling operations, *i.e.*, max-pooling and average-pooling. As shown in Table 1, PDG with part-level aggregation improves PDG-max and PDG-

Table 3: Results (in %) of variants of PDG for shape classification on ModelNet to ScanobjectNN with different architectures.

| Method | Last five | Best |
|---|---|---|
| baseline | 59.8 | 61.5 |
| PDG (w/o CL) | 66.4 | 68.1 |
| PDG (w/o SCL) | 67.1 | 69.3 |
| PDG-max | 65.3 | 67.8 |
| PDG-avg | 65.4 | 69.0 |
| PDG-OT | 65.2 | 65.7 |
| PDG-16parts | 65.4 | 67.1 |
| **PDG** | **67.6** | **69.4** |

avg by more than 2.2%. This shows that our part-based feature aggregation module is a superior way to aggregate the aligned part-based features and brings a significant performance boost to our network. We finally examine the effects of the contrastive learning framework defined in Sect. 4. As shown in Table 3, with the contrastive learning framework, our method achieves better results than PDG (w/o CL) with 1.2% improvement. We also remove the shape-level contrastive loss in PDG, denoted as PDG (w/o SCL), which performs slightly worse than PDG by 0.5% (0.1%), while still outperforming baseline by 7.3% (7.8%). This proves the effectiveness of our contrastive learning framework to improve the robustness of the aligned part-based representations and the major performance gain of PDG is derived from the design of part-based feature representation.

**Selection the scale of part.** As discussed in Sect. 2, the scale of the part-level features will influence the generalization and discrimination abilities. Specifically, the discrimination ability of part-level features decreases when the scale reduces, while the generalization ability improves. We show the results of PDG with 8 parts (512 points in each part) and 16 parts (256 points in each part) in Table 3, PDG with 8 parts (i.e., PDG in Table 3) outperforms PDG-16parts by 2.2%. Thus we choose 8 parts with 512 points in each part for a better balance of discrimination and generalization abilities.

**Feature visualization and analysis.** We first visualize the learned shape-level and part-level features of PointNet [5] of class "chair" and "sofa" from source domain ($M$) and target domain ($S^\star$) in Fig. 4. We can observe that in Fig. 4 (a), the shape-level features in source domain display a high inter-class variance, while the features of "sofa" in target domain are closer to the features of "chair" in source domain due to the geometry shift between two domain data. This implies better discrimination ability of shape-level representation in source domain while worse in target domain. Conversely, the part-level features of distinct domains and classes cluster together as shown in Fig. 4 (c), confirming our idea that the local geometric structures represented by part-level features are shared across different domains and classes and could reduce geometry shift effectively. However, the less discriminative part-level features are hard to be aggregated into a global representation for classification. Our PGD solves this problem effectively as shown in Fig. 4 (b)(d). Our aligned part-based representations in Fig. 4 (d) improve the discrimination ability of part-level features while

keeping the generalization ability. And in Fig. 4 (b), we observe both lower inter-domain variance and higher inter-class variance in the global representations of PDG compared with shape-level features of PointNet in Fig. 4 (a), resulting in the improvement in classification accuracy (67.6% $vs.$ 59.8%).

# 6 Related Work

**Deep classification on point clouds.** The early work PointNet [5] processes points independently using MLP and aggregates them by a max-pooling to obtain the permutation invariant representation, ignoring the vital local geometric structure of point clouds. DGCNN [6] models point clouds as dynamical graphs and proposes a dynamic edge convolution for classification of point clouds. Succeeding methods [1–4, 6–10] achieve improved performance for 3D shape recognition. However, all these methods assume that the training and test data are from the same domain thus can be weak at generalizing learned representations to new domains. A method that could improve the domain generalization ability of point cloud representation is desirable.

**Domain Generalization and Adaptation for Point Cloud classification.** How to represent 3D point clouds in different domains is an old but unsolved problem. [44] is one of the first methods to explore the 3D domain adaptation problem, which leverages synthetic scans of 3D scenes from Google 3D Warehouse to train an object detection system for 3D point clouds real-scan data. [45] and [46] applied hough transform to the problem of robust 3D shape feature learning and evaluated on point cloud classification and retrieval tasks of 3D domain generalization problem. [47] factored low-dimensional deformations and pose variations of the 3d shapes and recognized them in the scanned cluttered indoor scene, which is also a 3D domain generalization problem. [48] focused on the CAD-to-scan retrieval task of the 3D domain generalization problem and introduced a method called CAD-Deform to obtain accurate CAD-to-scan fits by non-rigidly deforming retrieved CAD models. Recently, domain adaptation or generalization problem on deep point cloud classification network [22, 25, 40–42, 49–52] has drawn more attention. PointDAN [25] proposes a 3D domain adaptation network for point cloud classification by minimizing an MMD loss to align local features across domains. [40–42, 49–52] further improve the results by designing a series of self-training tasks. However, all these methods are designed for 3D domain adaptation with known target domain data distribution. MetaSets [22] explores 3D domain generalization problem that the target domain data is inaccessible when training, and proposes to meta-learn point cloud representations from a group of point cloud transformations. Compared with them, we tackle the domain generalization problem by reducing geometry shifts using aligned part-based representation. Experiments on 3DDA and 3DDG benchmarks show that our PDG outperforms Metasets method by a large margin.

**Contrastive self-supervised learning.** Contrastive learning [53] which learns representation by contrasting positive pairs and negative pairs has achieved increasingly success for self-supervised learning [35, 54–59]. The learned representation could capture information shared between different augmentations of the same input. In this paper, our motivation is that aligned part-based representation should be similar under different local point transformations. To this end, we propose a contrastive learning framework upon part-based shape representation and improve the robustness of aligned part-based representation effectively.

# 7 Conclusion

In this work, we propose a novel part-based domain generalization network for 3D point cloud classification. We propose to learn generalizable part-based feature representations of point clouds to reduce the geometry gap of distinct domains. A contrastive learning framework is further designed to improve the robustness of the part-based feature representation. Extensive experiments justify its effectiveness. The current method is designed for point cloud classification, and we plan to extend it to point cloud segmentation or detection in the future work.

**Acknowledgment** This work was supported by NSFC with grant numbers of 11971373, 12125104, U20B2075, 12090021, 61721002.

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
