# Learning Generalizable Part-based Feature Representation for 3D Point Clouds (Supplementary Material)

**Xin Wei, Xiang Gu, Jian Sun** (✉)
School of Mathematics and Statistics, Xi'an Jiaotong University, P.R. China
{wxmath, xianggu}@stu.xjtu.edu.cn, jiansun@xjtu.edu.cn

## A. Sensitivity analysis on loss coefficients $\lambda_p$ and $\lambda_C$

As defined in Eqn.(8), the overall training loss of PDG consists of the shape loss $\mathcal{L}_{shape}$, part loss $\mathcal{L}_{part}$, shape-level contrastive loss $\mathcal{L}_{shape}^C$, and part-level contrastive loss $\mathcal{L}_{part}^C$, where the coefficients $\lambda_p$ and $\lambda_C$ control the weighting factors between these loss functions. We conduct experiments on $M \to S^\star$ task to examine how $\lambda_p$ and $\lambda_C$ could affect the performance. For the experiments on $\lambda_p$, we fix $\lambda_C$ to 0.01. And for the experiments on $\lambda_C$, $\lambda_p$ is fixed to 0.05. As shown in Fig. S-1 (a), the performance of PDG is not sensitive to $\lambda_p$ in the range of [0.01, 0.09], where the standard deviation of nine results is 0.71. And in Fig. S-1 (b), the standard deviation of nine classification results is 0.66, demonstrating that our PDG is also not sensitive to $\lambda_C$ in the range of [0.006, 0.05].

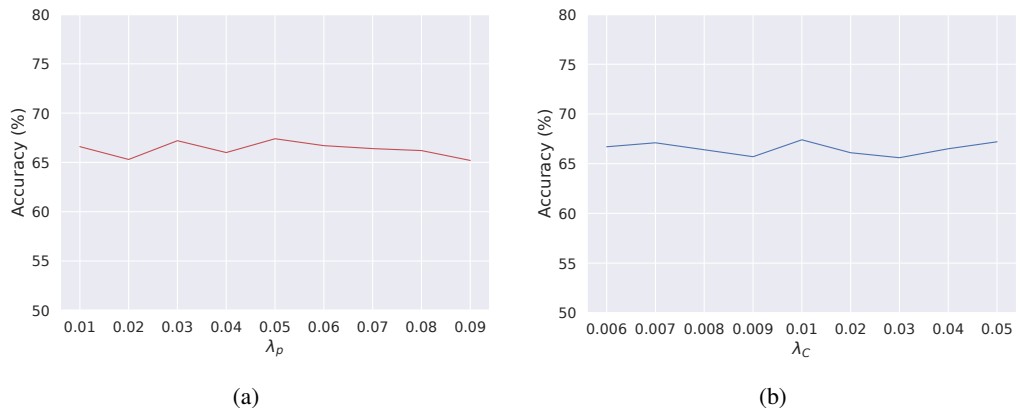

Figure S-1: Sensitivity analysis about $\lambda_p$ and $\lambda_C$ on $M \to S^\star$ task. (a) $\lambda_p$ is set in the range of [0.01, 0.09] and $\lambda_C$ is fixed to 0.01. (b) $\lambda_C$ is set in the range of [0.006, 0.05] and $\lambda_p$ is fixed to 0.05.

## B. Test error curves and training losses

We show the test error curves of PointNet, MetaSets (PointNet), and PDG (PointNet) in target domain data on task $M \to S^\star$ as shown in Fig. S-2 (a). We can observe that PointNet achieves relatively high test error and MetaSets (PointNet) could not sufficiently decrease the test errors. Compared with them, the test error of our proposed PDG (PointNet) decreases to a low value. Figure S-2 (b) shows all training losses of PDG (PointNet) including shape loss $\mathcal{L}_{shape}$, part loss $\mathcal{L}_{part}$, shape-level

36th Conference on Neural Information Processing Systems (NeurIPS 2022).

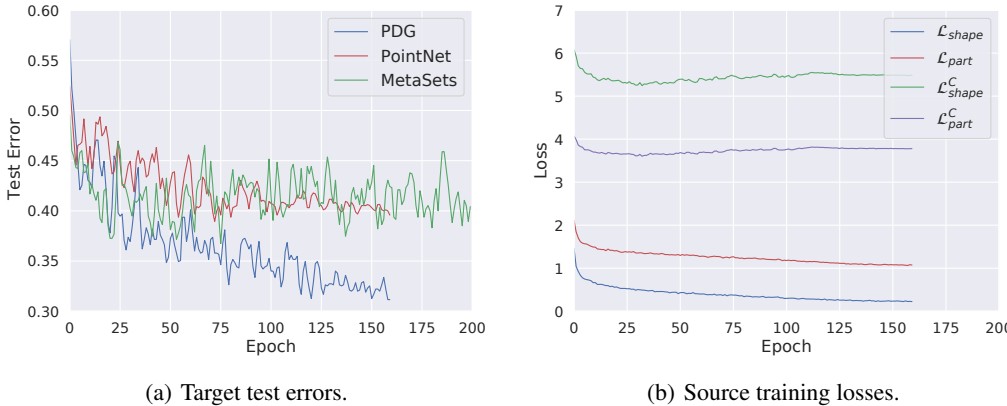

(a) Target test errors.        (b) Source training losses.

Figure S-2: Test error curves of different methods and training losses of PDG on $M \rightarrow S^{\star}$ task. (a) Test error curves of PointNet, MetaSets (PointNet), and PDG (PointNet) in target domain during network training. (b) Training losses of PDG (PointNet) in source domain during network optimization.

contrastive loss $\mathcal{L}_{shape}^{C}$, and part-level contrastive loss $\mathcal{L}_{part}^{C}$. All these losses are optimized stably during network training.

## C. Time and space complexity

Table S-1 summaries the comparisons of time and space cost of PointNet, PDG (PointNet) and MetaSets (PointNet). All experiments are conducted on a single Tesla V100 GPU with batch-size 32. We can observe that PDG (PointNet) achieves the best balance on performance and computation cost. Compared with MetaSets (PointNet), our proposed PDG (PointNet) improves its classification accuracy by 7.3% while taking only respectively 3.6% and 86.7% time and space cost of MetaSets (PointNet).

Table S-1: Time and memory cost of different models.

|  | PointNet | MetaSets (PointNet) | PDG (PointNet) |
|---|---|---|---|
| Time per epoch (s) | 2.94 | 238.64 | 8.76 |
| Total training time (m) | 7.67 | 763.46 | 24.15 |
| GPU memory (MB) | 2953 | 5097 | 4423 |
| Accuracy (%) | 59.8 | 60.3 | 67.6 |

## D. A comparison of typical shapes from different domains

In Sect. 5, we evaluate different methods on 3DDG benchmarks consisting of three domains: Model-Net, ShapeNet, and ScanObjectNN. We show some typical shapes in each class from three datasets in Fig. S-3. It is obvious that CAD shapes from synthetic datasets (ModelNet and ShapeNet) are cleaner, while shapes from real scanned dataset (ScanobjectNN) suffer from the background occurrence, object partiality, and various deformation variants. These geometry variances cause the domain gap and degrade the performance of 3D classification models when generalizing to an unseen domain.

## E. Visualization of part-template features

As discussed in Sect. 3.2, we construct a part-template feature space $H$ and each part-template feature could encode part-level geometry. We illustrate some learned part-template features in Fig. S-4 and Fig. S-5. We train a PDG (DGCNN) on ModelNet dataset and extract part-level features from source

domain (ModelNet dataset) and target domain (ScanObjectNN dataset). Given a part-template feature as a query, part-level features from source and target domains are retrieved respectively. In each row of Fig. S-4, we display 8 parts from source domain (in blue box), and 8 parts from target domain (in red box) whose features are most similar to the same part-template feature in respective domain. As shown in Fig. S-4 and Fig. S-5, part-template features could encode various local geometric structures in source domain and provide references for target domain.

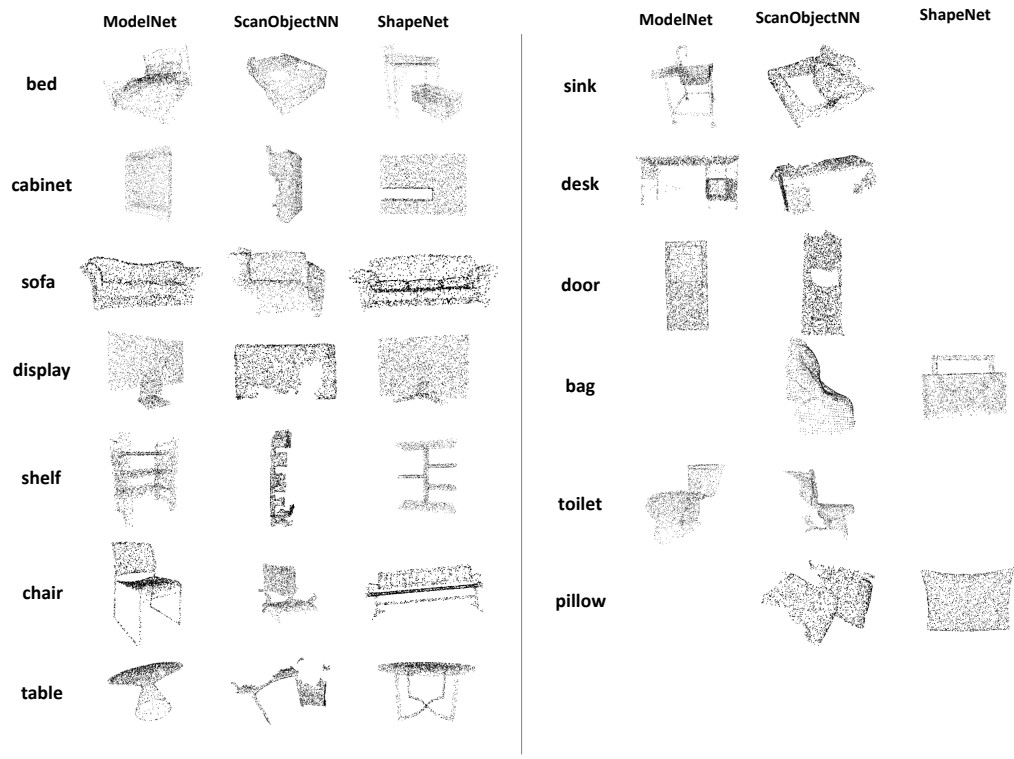

Figure S-3: Samples from ModelNet dataset, ShapeNet dataset, and ScanObjectNN dataset. ModelNet and ScanObjectNN have 11 shared classes while ShapeNet and ScanObjetNN have 9 shared classes.

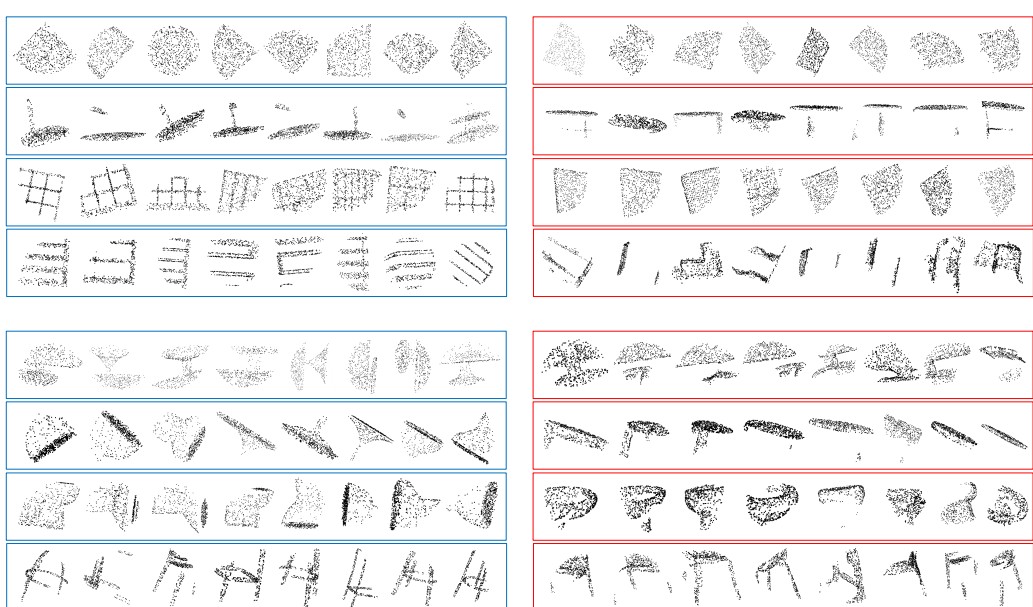

Figure S-4: Visualization of part-template features. In each row we display 8 parts retrieved from source domain (in blue box) and 8 parts from target domain (in red box) whose part-level features are most similar to the same part-template feature in respective domain.

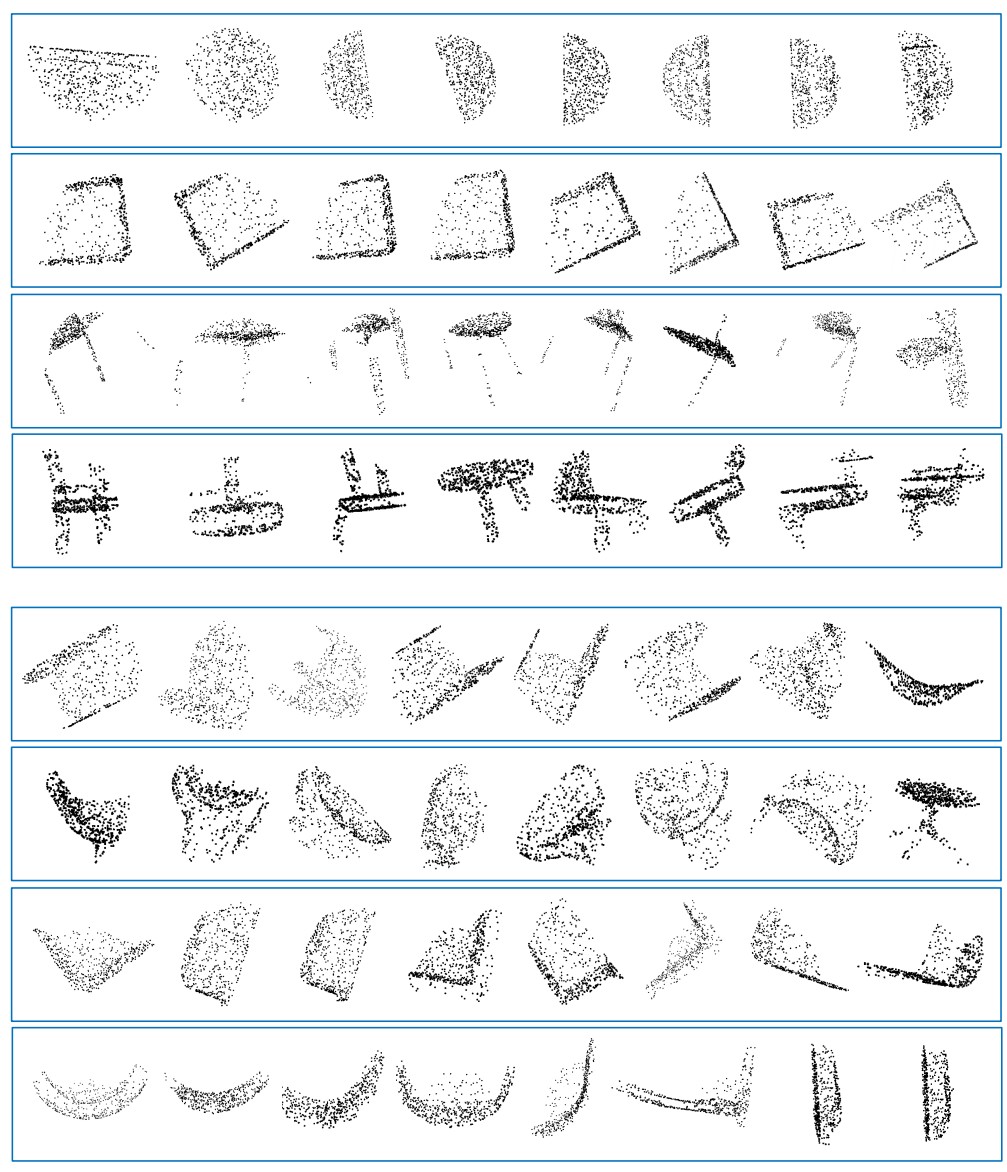

Figure S-5: Examples of part-template features. For each part-template feature, we display 8 parts retrieved from source domain whose part-level feature are most similar to the part-template features in each row.