# OpenReview forum: "Learning Generalizable Part-based Feature Representation for 3D Point Clouds"
_NeurIPS.cc/2022/Conference — NeurIPS 2022 Accept_

### Official Review · Reviewer_GcXJ · 2022-07-09

**Rating:** 6
**Confidence:** 3
**Soundness:** 3 good
**Presentation:** 3 good
**Contribution:** 2 fair

**Summary:**

The presented method detects “global” features (PointNet or DGCNN)  locally on sampled points. Then learn relations between those local representations as part-level aggregation. The performance is further improved by contrastive learning.

Authors evaluate the approach on several cross domain datasets where the method is learnt on one domain and tested on another one. The target (test) domain is inaccessible during training.

Domain adaptation is an important and well old problem for 3d point cloud processing.


**Questions:**

-Why did authors not report M->S like a comparison to other methods? It would be a very nice comparison to other papers [Achituve21, Shen22,..]. It looks strange that evaluation is done with datasets [11],[12],[14], but popular [13] (ScanNet) is missing, ScanNet is often used in previous methods  [Achituve21, Shen22,..]. Readers can think about many reasons why authors didn't report it and it looks strange that authors perform too many results, but the most important settings are somehow missing.

-Authors say that they executed the competitors' code, still, they have different performance values than in other papers (in other papers they also vary from paper to paper).  E.g. M->S*, MetaSet paper claims they have 72% (so better than the presented method), but authors report ~66% as best for Metaset. Is it a problem of the field that once you download a repository of a paper you obtain different results that were originally published? The difference is higher than the paper’s improvement :(


-If one looks on the proposed method as using "global" features (e.g. PointNet) locally on parts of the object and then learning relations of those local responses, then this is an old method how to represent 3d point cloud object in the scene between different domains, for example [Woodford13, Knopp10] learns spatial configuration of such features for the nice 3d shapes and apply it on the 3d scans, [Kim12] learns parts deformation of the 3d shapes and recognize them in the scanned cluttered scene. [Lai10] Looks like one of the first method to do 3d point domain adaption. Here is list of related missing work:
[Lai10] Lai and Fox, Object Recognition in 3D Point Clouds Using Web Data and Domain Adaptation, IJRR 2010
[Ishimtse20] Ishimtse at all, CAD-Deform: Deformable Fitting of CAD Models to 3D Scans, ECCV 2020
[Qi19] Qi et all, Deep Hough Voting for 3D Object Detection in Point Clouds, ICCV 2019
[Golovinskiy09] Golovinskiy et al, Shape-based Recognition of 3D Point Clouds in Urban Environments, ICCV'09
[Knopp10] Knopp et all; Hough Transform and 3D SURF for Robust Three Dimensional Classification, ECCV 2010
[Woodford13] Woodford et all, Demisting the Hough Transform for 3D Shape Recognition and Registration, IJSV 2013
[Kim12] Y.M. Kim et al., Acquiring 3D Indoor Environments with Variability and Repetition, ACM ToG, SIGGRAPH Asia 2012


**Limitations:**

It looks there is no potential for negative social impact of the work.


**Strengths And Weaknesses:**

Pros:
-Clear motivation, I like the motivation by features distance between domains (Fig. 2)


-While the idea of learning part-based models from local features is old and highly researched, the presented method on 3D point clouds with Neural Networks focused for the domain adaptations seems novel.

-The problem of domain adaptation of 3d point cloud processing when the target domain is unavailable  during training is a very important and unsolved problem.

-Most of questions I had during reading were further answered


Cons:
-The approach is motivated from many previous works that focus on domain adaptation for 3d point cloud that are not cited, though the approach is novel by using it in neural networks.

-Authors did not evaluate on some datasets pairs (training-testing) that will allow much broader comparison. That raises several questions why. I believe it will be also good to report why presented numbers of other methods differ from original papers.

---

> ### Author Response · Authors · 2022-08-02
> **Response to Reviewer GcXJ part (2/2)**
>
> **Q3: More related works on 3D domain adaptation.**
>
> Thanks for this suggestion. How to represent 3D point clouds in different domains is an old but unsolved problem, which has drawn more attention recently. [R3] is one of the first methods to explore the 3D domain adaptation problem, which leverages synthetic scans of 3D scenes from Google 3D Warehouse to train an object detection system for 3D point clouds real-scan data. [R4] and [R5] applied hough transform to the problem of robust 3D shape feature learning and evaluated on point cloud classification and retrieval tasks of 3D domain generalization problem. [R6] factored low-dimensional deformations and pose variations of the 3d shapes and recognized them in the scanned cluttered indoor scene, which is also a 3D domain generalization problem. [R7] focused on the CAD-to-scan retrieval task of the 3D domain generalization problem and introduced a method called CAD-Deform to obtain accurate CAD-to-scan fits by non-rigidly deforming retrieved CAD models. [R8] and [R9] only concentrated on single domain problems. Compared with them, our proposed PDG method works on 3D domain generalization problem, and is built based on the proposed part-based deep feature learning approach. As suggested, we will cite and include these related works in sections of introduction / related works.
>
> [R1] Ganin, Yaroslav, et al. Domain-adversarial training of neural networks. JMLR, 2016.
>
> [R2] Jonathan Sauder and Bjarne Sievers. Self-supervised deep learning on point clouds by reconstructing space. In Neurips, 2019.
>
> [R3] Lai, Kevin, and Dieter Fox. Object recognition in 3D point clouds using web data and domain adaptation. In IJRR, 2010.
>
> [R4] Knopp, Jan, et al. Hough transform and 3D SURF for robust three dimensional classification. In ECCV, 2010.
>
> [R5] Woodford, Oliver J., et al. Demisting the Hough transform for 3D shape recognition and registration. IJCV, 2013.
>
> [R6] Kim, Young Min, et al. Acquiring 3d indoor environments with variability and repetition. ACM ToG, 2012.
>
> [R7] Ishimtsev, Vladislav, et al. Cad-deform: Deformable fitting of cad models to 3d scans. In ECCV, 2020.
>
> [R8] Golovinskiy, Aleksey, et al. Shape-based recognition of 3D point clouds in urban environments. In ICCV, 2009.
>
> [R9] Qi, Charles R., et al. Deep hough voting for 3d object detection in point clouds. In ICCV, 2019.

---

> ### Author Response · Authors · 2022-08-02
> **Response to Reviewer GcXJ part (1/2)**
>
> We thank the reviewer for the comments and suggestions. We will revise our paper accordingly.
>
> **Q1:  Results on PointDA-10 dataset.**
>
> For a fair comparison with current 3DDG method (MetaSets), we only conduct experiments on the dataset provided by them. We evaluate our method on PointDA-10 dataset [38] and compare it with current 3D domain adaptation methods which use target domain data in training procedure (denoted as DA in Table r3-1) and 3DDG methods where target domain data is inaccessible during training (denoted as DG). PointDA-10 dataset consists of three domains, i.e., synthetic dataset ModelNet-10 (**$M$**),  synthetic dataset ShapeNet-10 (**$S$**), and real scan dataset ScanNet-10 (**$S^{\star}$**). Six point cloud domain generalization tasks are built, including $M\to S$,  $M\to S^{\star}$, $S\to M$, $S\to S^{\star}$, $S^{\star}\to M$, $S^{\star}\to S$. These tasks include all three settings, i.e., real-to-synthetic, synthetic-to-real, and synthetic-to-synthetic. We use DGCNN as the backbone like previous methods and the same training setting as in 3DDG benchmark. The results are shown in Table r3-1. For MetaSets and PDG, we report the "Last five" results.
>
> Table r3-1. Classification accuracy (in %) of various 3DDA and 3DDG methods on PointDA-10 dataset. Results of methods which do not use target data during training are in bold. Note that the results of DA and DG methods cannot be fairly compared because DA methods use target domain data in training.
>
> | **Method**           | **DA/DG** | **$M\to S$** | **$M\to S^{\star}$** | **$S\to M$** | **$S\to S^{\star}$** | **$S^{\star}\to M$** | **$S^{\star}\to S$** | **Average** |
> | :------------------- | :-------: | :----------: | :------------: | :----------: | :------------: | :------------: | :------------: | :---------: |
> | Supervised           |     -     |     93.9     |      78.4      |     96.2     |      78.4      |      96.2      |      93.9      |    89.5     |
> | **w / o Adapt** | **-** | **83.3** | **43.8** | **75.5** | **42.5** | **63.8** | **64.2** | **62.2** |
> | DANN [R1]            |    DA     |     74.8     |      42.1      |     57.5     |      50.9      |      43.7      |      71.6      |    56.8     |
> | PointDAN [38]        |    DA     |     83.9     |      44.8      |     63.3     |      45.7      |      43.6      |      56.4      |    56.3     |
> | RS [R2]              |    DA     |     79.9     |      46.7      |     75.2     |      51.4      |      71.8      |      71.2      |    66.0     |
> | DefRec + PCM [39]    |    DA     |     81.7     |      51.8      |     78.6     |      54.5      |      73.7      |      71.1      |    68.6     |
> | GAST [40]            |    DA     |     84.8     |      59.8      |     80.8     |      56.7      |      81.1      |      74.9      |    73.0     |
> | **MetaSets [22]** | **DG** | **86.0** | **52.3** | **67.3** | **42.1** | **69.8** | **69.5** | **64.5** |
> | **PDG (Ours)** | **DG** | **85.6** | **57.9** | **73.1** | **50.0** | **70.3** | **66.3** | **67.2** |
>
> From Table r3-1,  PDG improves the baseline methods by **5.0%** in the average accuracy of all tasks and outperforms 3DDG method Metasets by **2.7%**. Specifically, PDG beats MetaSets in four tasks $M\to S^{\star}$, $S\to M$, $S\to S^{\star}$, and $S^{\star}\to M$. For two synthetic-to-real tasks $M\to S^{\star}$ and $S\to S^{\star}​$, the improvements are **5.6%** and **7.9%**. We can also find that PDG exceeds some 3DDA methods including DANN [R1], PointDAN [38], and RS [R2]. We will properly cite more 3D domain adaptation works and include these results in the final version.
>
> **Q2: Results of MetaSets.**
>
> For all experiments, PDG and MetaSets use same datasets, data preparation and experiment setting. For all experiments with PointNet as backbone, we report their results by directly running their published code. They report 68.28%, 57.19%, 55.25%, and 49.50%  in the four tasks,  while we get 66.3%, 57.5%,  55.3%, and 50.3% as "Best" for MetaSets. These results are similar to those reported in their paper. As for the experiments with DGCNN as the backbone, the corresponding codes of MetaSets using DGCNN backbone is not provided. For fair comparison in the experiment settings, we use exactly the same implementation of DGCNN as backbones in our method and MetaSets by replacing the backbone by the same DGCNN in their source codes. The results of MetaSets with DGCNN backbone are reported in the above way which exactly matches our method in dataset, backbone, data augmentations, etc.

---

### Official Review · Reviewer_KRL8 · 2022-07-11

**Rating:** 6
**Confidence:** 5
**Soundness:** 3 good
**Presentation:** 3 good
**Contribution:** 3 good

**Summary:**

The authors present a new method for generalizing point cloud classification from synthetic to real data. The authors argue that the local geometric features are more generalizable than the whole shape. They focus on part-based feature representation, and design a part-based domain generalization network for the point cloud classification task. The key idea is to build a common feature space using a part template, and then align the part-level features of the source and the target domain to this template. The authors demonstrate that the proposed method achieves state-of-the-art performance on the 3DDG benchmark.

**Questions:**

(1) For completeness, I expect to find some information about how the part template is defined. However, it seems not provided except that in Line 246 the authors mentioned that they followed the experiment setting in MetaSets [22]. Could you clarify how the parts are defined? How does this affect the final performance as the proposed method is largely based on part features?

(2) While the benchmarks are built upon ModelNet/ShapeNet and ScanObjectNN, I see that not all data are utilized. For example, the experiments are only done on two basic variants of ScanObjectNN including OBJ-ONLY and OBJ-BG. How does the transfer work if the target is the hardest variant of the ScanObjectNN?

(3) As a domain generalization problem, it seems insufficient to conduct only sim-to-real transfer experiment. How about real-to-sim, i.e., training on ScanObjectNN and testing on ModelNet/ShapeNet? In practice, this could be used to build applications such as retrieving a CAD model from a given scan. As all data is available, I wish to further understand what challenges remain that make this experiment not conducted in both MetaSets and this paper.


**Limitations:**

I think the related work section in this paper is quite short and needs some revisions. First, while not exactly the same, I found in the literature there are some 3D tasks that link different domains together such as scans and CAD object retrieval. I think this is worth some further discussions about the connections of these specific tasks with the domain generalization problem presented in this paper.
[A] SHREC'17: RGB-D to CAD retrieval with ObjectNN dataset, 2017, 2018.

**Strengths And Weaknesses:**

- I like the idea to align local geometric features to solve domain generalization on point clouds. This idea is novel and significant. The technical approach to implement this idea is sound, and the experimental results demonstrate good performance.

- I also like the idea to verify the hypothesis that local geometric features are more generalizable than global features in Fig. 2. However, I would like to point out a few issues here.
  (1) It is true that in general reducing the part size leads to better generalization. But where is the limit? At the very least each part can be reduced to a point, but I do not believe that point-based features are the most generalizable. It could be more interesting to identify by how many points per part we would reach to limit of generalization here.
  (2) 512-part-level and 256-part-level mean 512 and 256 points per part, respectively I guess. This sounds confusing as I can also think of it as 512 parts and 256 parts. It is better to revise this wording, like 512-points-per-part and 256-points-per-part.

- I also value the clarity of the writing, which is very nice and easy to read.

- Despite its great values, the paper suffers from the following issues.
  (1) In terms of technical approach, the contrastive learning part is less well connected to the part-based features for domain generalization. For example, if the authors wish to use contrastive learning, at least shape-level contrastive loss should be used for the baseline methods as well. Or the comparisons should be separated with a table with no contrastive learning utilized. In Table 1, as I understand, the baselines are without contrastive loss but the PDG is with contrastive loss. Please correct me if I misunderstood.
  (2) My second concern is that the experiments conducted are somewhat simplistic. I expect deeper analysis and more experimental settings to be done. Please see my comments in the question section.

---

> ### Author Response · Authors · 2022-08-02
> **Response to Reviewer KRL8 part (4/4)**
>
> **Q6: Related works on scan-to-CAD shape retrieval.**
>
> We have conducted several real-to-synthetic DG classification tasks and show the effectiveness of our approach, as shown in Q5. The rgb-d real scan to CAD retrieval tasks [R3, R4, R5] aims to retrieve a similar CAD model to a given query real scan 3D object. It relies on generalizable feature representation and the similarity measure of real scan objects and CAD objects. We will properly cite and discuss these works in the section on related work. Though our experiments mainly work on domain generalization classification tasks, our part-based feature representation can be possibly extended to the rgb-d real scan to CAD retrieval task, deserving us to try in future work. This can be possibly achieved as follows. Given query shape and candidate shape, they can be represented by two sets of part-based features and the similarity can be calculated by a set-to-set measure. Considering the situation that scanned shapes often suffer from object partiality and different deformation variants. Part-based feature representations are suitable for cross-domain 3D shape retrieval. We will include this future work in the conclusion section.
>
> [R1] Ganin, Yaroslav, et al. Domain-adversarial training of neural networks. JMLR, 2016.
>
> [R2] Jonathan Sauder and Bjarne Sievers. Self-supervised deep learning on point clouds by reconstructing space. In Neurips, 2019.
>
> [R3] Hua, Binh Son, et al. SHREC'17: RGB-D to CAD retrieval with ObjectNN dataset.
>
> [R4] Pham, Quang-Hieu, et al. SHREC’18: Rgb-d object-to-cad retrieval.
>
> [R5] Dahnert, Manuel, et al. Joint embedding of 3d scan and cad objects. In ICCV, 2019.

---

> ### Author Response · Authors · 2022-08-02
> **Response to Reviewer KRL8 part (3/4)**
>
> **Q5: Real-to-synthetic setting.**
>
> We evaluate PointNet, MetaSets (PointNet) and PDG (PointNet) on the tasks of  $S^{\star}\to M$  and $S^{\star}\to S$, where models are trained on real scan dataset ScanObjectNN (**$S^{\star}$**) and test on synthetic datasets ModelNet (**$M$**) and ShapeNet (**$S$**). Results are shown in Table R2-4. As shown in Table r2-4, both MetaSets and PDG improve baseline in two real-to-synthetic settings.
>
> Table r2-4. Classification results (in %) of PointNet, MetaSets (PointNet) and PDG (PointNet) on the tasks of  $S^{\star}\to M$ and $S^{\star}\to S$.
>
> | **Method**          | **$S^{\star}\to M$** | **$S^{\star}\to S$** |
> | ------------------- | :--------------: | :--------------: |
> | PointNet            |   63.7 (71.0)    |   74.7 (80.2)    |
> | MetaSets (PointNet) |   64.3 (71.9)    | **77.0** (81.2)  |
> | PDG (PointNet)      | **66.7 (72.5)**  | 76.1 (**81.4**)  |
>
> We also conduct experiments on PointDA-10 dataset which consists of three domains, i.e., synthetic dataset ModelNet-10 (**$M$**),  synthetic dataset ShapeNet-10 (**$S$**), and real scan dataset ScanNet-10. Six point cloud domain generalization tasks are built, including $M\to S$, $M\to S^{\star}$, $S\to M$, $S\to S^{\star}$, $S^{\star}\to M$, $S^{\star}\to S$. We use DGCNN as the backbone like previous methods and the same training setting as in 3DDG benchmark. The results are shown in Table r3-1. For MetaSets and PDG, we report the "Last five" results. These tasks include all three settings, i.e., real-to-synthetic, synthetic-to-real, and synthetic-to-synthetic.
>
> Table r2-5. Classification accuracy (in %) of various 3DDA and 3DDG methods on PointDA-10 dataset. Results of methods which do not use target domain data during training are in bold. Note that the results of DA and DG methods cannot be fairly compared because DA methods use target domain data in training.
>
> | **Method**           | **DA/DG** | **$M\to S$** | **$M\to S^{\star}$** | **$S\to M$** | **$S\to S^{\star}$** | **$S^{\star}\to M$** | **$S^{\star}\to S$** | **Average** |
> | :------------------- | :-------: | :----------: | :------------: | :----------: | :------------: | :------------: | :------------: | :---------: |
> | Supervised           |     -     |     93.9     |      78.4      |     96.2     |      78.4      |      96.2      |      93.9      |    89.5     |
> | **w / o Adapt** | **-** | **83.3** | **43.8** | **75.5** | **42.5** | **63.8** | **64.2** | **62.2** |
> | DANN [R1]            |    DA     |     74.8     |      42.1      |     57.5     |      50.9      |      43.7      |      71.6      |    56.8     |
> | PointDAN [38]        |    DA     |     83.9     |      44.8      |     63.3     |      45.7      |      43.6      |      56.4      |    56.3     |
> | RS [R2]              |    DA     |     79.9     |      46.7      |     75.2     |      51.4      |      71.8      |      71.2      |    66.0     |
> | DefRec + PCM [39]    |    DA     |     81.7     |      51.8      |     78.6     |      54.5      |      73.7      |      71.1      |    68.6     |
> | GAST [40]            |    DA     |     84.8     |      59.8      |     80.8     |      56.7      |      81.1      |      74.9      |    73.0     |
> | **MetaSets [22]** | **DG** | **86.0** | **52.3** | **67.3** | **42.1** | **69.8** | **69.5** | **64.5** |
> | **PDG (Ours)** | **DG** | **85.6** | **57.9** | **73.1** | **50.0** | **70.3** | **66.3** | **67.2** |
>
>
>
> In Table r2-5, PDG improves baseline by **6.5%** and **2.1%** in two real-to-synthetic tasks, i.e., $S^{\star}\to M$ and $S^{\star}\to S$. Compared with 3DDG method MetaSets, PDG performs better in the $S^{\star}\to M$ task and worse in $S^{\star}\to S$ task. Considering the average performance in all tasks, PDG outperforms baseline method by **5.0%** and MetaSets by **2.7%**.  It is noticeable that PDG even exceeds some 3DDA methods including DANN [R1], PointDAN [38], and RS [R2]. These results demonstrate that PDG could also solve the real-to-synthetic point cloud generalization problem.

---

> ### Author Response · Authors · 2022-08-02
> **Response to Reviewer KRL8 part (2/4)**
>
> **Q3: Ablation on shape-level contrastive learning loss.**
>
> In this paper, we build a contrastive learning framework upon part-based shape representation to improve the robustness of the learned aligned part-based features. Specifically, part-level contrastive loss encourages learned part-based feature of a part of a shape under different local transformations to be consistent in the feature space in an unsupervised manner, while shape-level contrastive loss pushes global representation of shapes in the same class together. The shape-level contrastive loss could be used for the baseline method. For a fair comparison with the baseline method, we remove the shape-level contrastive learning loss in PDG, denoted as PDG (w/o SCL) and the results of four domain generalization tasks (**$M\to S^{\star}$**, **$M\to S^{\star}_B$**, **$ S\to S^{\star}$**, **$S\to S^{\star}_B$**) are presented in Table r2-2, where **$M$**, **$S$**, **$S^{\star}$**, **$S^{\star}_B$** represented ModelNet, ShapeNet, ScanObjetNN, ScanObjectNN with background respectively.
>
> Table r2-2. Ablation on shape-level contrastive loss.
>
> | **Method**          | **$M\to S^{\star}$**  | **$M\to S^{\star}_B$** | **$ S\to S^{\star}$** | **$S\to S^{\star}_B$** |     **Average**     |
> | :------------------ | :-------------: | :--------------: | :-------------: | :--------------: | :-------------: |
> | PointNet            |   59.8 (61.5)   |   51.5 (53.8)    |   55.9 (57.4)   |   51.0 (54.0)    |   54.5 (56.7)   |
> | MetaSets (PointNet) |   60.3 (66.3)   |   52.4 (57.5)    |   51.8 (55.3)   |   44.3 (50.3)    |   52.2 (57.0)   |
> | PDG (PointNet)      | **67.6 (69.4)** | **58.5 (61.1)**  | **57.3 (61.8)** | **51.3 (55.5)**  | **58.7 (62.0)** |
> | PDG (w/o SCL)       |   67.1 (69.3)   |   57.1 (60.0)    |   57.1 (60.1)   |   51.1 (54.8)    |   58.1 (61.1)   |
>
> As shown in Table. r2-2,  PDG (w/o SCL) performs slightly worse than PDG by **0.6%** in average accuracy, while still outperforms baseline and MetaSets by **3.6%** and **5.9%** respectively. These results demonstrate that the major performance gain of PDG is derived by the design of part-based feature representation.
>
> **Q4: Generalization to hardest version of ScanObjectNN.**
>
> We evaluate PointNet, MetaSets (PointNet) and PDG (PointNet) on the tasks of  $M\to S^{\star}_H$ and $S\to S^{\star}_H$ , where models are trained on ModelNet (**$M$**) and ShapeNet (**$S$**) respectively, then tested on the hardest version of ScanObjectNN (**$S^{\star}_H$**). Results are in Table R2-3.
>
> Table r2-3. Classification results (in %) of PointNet, MetaSets (PointNet) and PDG (PointNet) on the task of $M\to S^{\star}_H$ and $S\to S^{\star}_H$.
>
> | **Method**          | **$M\to S^{\star}_H$** | **$S\to S^{\star}_H$** |
> | ------------------- | :----------------: | :----------------: |
> | PointNet            |    50.0 (51.9)     |    49.0 (50.9)     |
> | MetaSets (PointNet) |    47.4 (54.4)     |    45.6 (50.0)     |
> | PDG (PointNet)      |  **56.6 (57.2)**   |  **51.3 (53.9)**   |
>
> In Table r2-3, it can be observed that PDG (PointNet) outperforms both PointNet and MetaSets (PointNet) in two tasks, which demonstrates that part-based feature representation learned by PDG are more generalizable to the shapes under large perturbations. We will include these results in the main paper.

---

> > ### Comment · Reviewer_KRL8 · 2022-08-05
> > **Thank you for the extra ablation study**
> >
> > Thank you for the details in the rebuttal. In my understanding, the contrastive learning part seems optional. Would it be better to shift section 4 of the paper to supplementary, and use the space to present such extra results of real-to-sim, the hardest variant of ScanObjectNN? As without the contrastive learning loss the model can outperform previous works then I do not see a reason to have it here.
> >
> > For the experiments in the rebuttal, please kindly provide DGCNN backbone in the revised version as well.
> >
> > I am generally fine with the responses and I'll increase my score accordingly.

---

> > > ### Author Response · Authors · 2022-08-05
> > > **Response to Reviewer KRL8**
> > >
> > > Thanks for the suggestions. We will consider to clarify the role of shape-level contrastive learning part, compress or move details in sect.4 to supplementary. Since we are allowed to have one extra page if the paper is accepted, we will include all these discussions in the paper. The results and discussions of our method and compared methods with PointNet and DGCNN as the backbone on real-to-sim setting and the hardest variant of ScanObjectNN setting will be included.

---

> ### Author Response · Authors · 2022-08-02
> **Response to Reviewer KRL8 part (1/4)**
>
> We thank the reviewer for the comments and suggestions. We will revise our paper accordingly.
>
> **Q1: Definition of part-template and part-based features.**
>
> As discussed in Sect.3.2, our part-template features are defined as a set of learnable ​$d$-dimension vectors ​${\\{H_i\\}}^{N_H} \in
>  R^{N_H \times d}$ for encoding the local geometric priors, where $d$ is the dimension of point-wise features depends on backbone network and $N_H$ is set to 384 as presented in the section of implementation details. The part-level features are defined in Sect.3.1. Given a point cloud ​$P=\\{x_1,x_2, ...,x_N\\}\in R^{N\times3}$ and corresponding point-wise features ​$Z=\\{z_1, z_2, ..., z_N\\}\in R^{N\times d}$, we represent it as a union of ​$M$ overlapped parts ​$P=Q_1\bigcup Q_2\bigcup ...\bigcup Q_M$, where each part ​$Q_i=\\{x_{i1}, ..., x_{ik}\\}\in R^{k\times 3}$ is defined as a center point ​$x_{i1}$ with its ​$k$ nearest neighbor points ​$\\{x_{i1}, ..., x_{ik}\\}$. $M$ center points are sampled by FPS for better coverage of the entire point cloud. The corresponding part-level feature ​$Z_{Q_i}\in R^d$ of part ​$Q_i$ is derived by max-pooling on point-wise features of each points in part ​$Q_i$. ​$M$ and ​$k$ are set to 8 and 512 as represented in the section of implementation details. We will further clarify these definitions in the revised paper.
>
> **Q2: The balance of generalization ability of part-level features and final performance of PDG.**
>
> As discussed in Sect.2, the balance of generalization ability and discrimination ability of part-level features depends on the part size. We explore how could point number per part influence the generalization ability of part-level features and the final performance of PDG. We conduct experiments on the task of  **$S\to S^{\star}$**, where models are trained on ModelNet (**$M$**) and tested on ScanObjectNN (**$S^{\star}$**).  For all results in this response, we report the "Last five" results of each methods and "Best" results are also provided in corresponding brackets. The generalization ability of part-level feature is evaluated by average A-distance in all classes, which is a measure to evaluate distribution discrepancy as referred to in Sect.2.
>
> Table r2-1. Average A-distance in all classes and classification accuracy (in %) under task $S\to S^{\star}$ for PDG with different part sizes.
>
> | **Part-size**   | **A-distance** | **$S\to S^{\star}$** |
> | --------------- | :----------------: | :--------------: |
> | 2048 (baseline) |        1.44        |   59.8 (61.5)    |
> | 512             |        1.06        | **67.6 (69.4)**  |
> | 256             |        0.90        |   65.4 (67.1)    |
> | 128             |        0.89        |   62.4 (65.3)    |
> | 64              |        0.88        |   63.1 (65.9)    |
> | 32              |        0.80        |   63.4 (67.6)    |
> | 16              |        0.72        |   64.4 (66.7)    |
> | 8               |      **0.68**      |   64.1 (65.4)    |
> | 1               |      **0.68**      |   62.3 (63.7)    |
>
> As shown in Table r2-1, the average A-distance drops quickly by 0.38 when part size decreases from 2048 to 512 and 0.16 when part size decreases from 512 to 256. If we further reduce the part size to 128, 64, 32, 16, and 8, the average A-distance drops slowly. When the part size reduces from 8 to 1, the average A-distance remains unchanged. The A-distance, as a measure of generalization ability of part-level features, reaches to limit when part size is 8.
>
> The part size also influences the performance of PDG for domain genearlization task. The best classification accuracy is achieved when part size is 512. With smaller part sizes, the performance of PDG drops due to the decreased discrimination ability of part-level features.
>
> We will revise 512-part-level and 256-part-level to 512-points-part-level and 256-points-part-level.

---

### Official Review · Reviewer_KHBY · 2022-07-22

**Rating:** 6
**Confidence:** 4
**Soundness:** 3 good
**Presentation:** 3 good
**Contribution:** 3 good

**Summary:**

The authors propose to utilize the part-level feature representation for the cross-domain generalization problem in point cloud applications. Given part-level features grouped from point-wise representations, the authors first align them to a learned feature dictionary via cross-attention, and then aligned-features are aggreged with a part-weighted maxpooling strategy. In addition, contrastive learning is conducted in both shape-level and part-level. Empirical results on standard DG benchmark datasets are presented for validation.

**Questions:**

See weaknesses.

**Limitations:**

Not found.

**Strengths And Weaknesses:**

Strengths:
1. The method is well motivated.  The authors find that part-level features present smaller distribution divergence than shape-level feature in the cross-domain tasks. Therefore, they propose to adopt part level features in the DG tasks.

2. Some interesting components are proposed and well justified. The proposed part-template features implicitly achieves the domain alignment by aligning both domains to the learned feature dictionary. The proposed part feature aggregation module outperfoms the popular max pooling module.

3. The proposed method achieves state-of-the-art performance on DG benchmarks.

Weaknesses:
1. I am wondering the relationship between the proposed part feature based DG method and the general point cloud models (e.g., PointNet++) that utilize part/local features.   For example, in the PointNet++, the part level feature is extracted and aggregated hierarchically, which is quite similar to the strategy adopted in this paper.  Could you clarify it?

2. Based on the first question, could the proposed module be adopted in general point cloud models? For example, could we replace the last max pooling layer of PointNet++ with the part feature aggregation module proposed in this paper?

3. As for the proposed techniques, the contrastive loss is widely adopted as the learning regularization and utilizing part level features is also a common practice. In my opinion, the main contribution is the implicit domain alignment with the learned feature dictionary and the part feature aggregation module. So I suggest that the authors include more related work on the application of dictionary learning in cross-domain problems, such as [1]

[1] Li, Shuai, et al. "Category dictionary guided unsupervised domain adaptation for object detection." Proceedings of the AAAI conference on artificial intelligence. Vol. 35. No. 3. 2021

---

> ### Author Response · Authors · 2022-08-02
> **Response to Reviewer KHBY**
>
> We thank the reviewer for the comments and suggestions. We will revise our paper accordingly.
>
> **Q1: Relationship between part-based domain generalization network and general point cloud models.**
>
> Exploiting local patterns has been proven to be important for the success of both CNN and point cloud convolutional networks. They progressively capture features at an increasingly larger receptive field. The ability to abstract local patterns guarantees the generalizability to unseen cases in a single domain. As discussed in Sect.3, we find that these local geometric structures are also shared across different shapes in distinct domains, while they are short of semantic information for classification. The balance of generalization ability and discrimination ability relies on the receptive field of the local pattern. Our part-level features could be regarded as the local features with a balanced receptive field.
>
> For general point cloud models, the local features are directly aggregated by a pooling operation. Compared with them, part-level features in PDG are aligned to the part-template features, resulting in part-based feature representations with better generalization ability. A part-based feature aggregation module finally aggregates these part-based feature representations to a global representation for each point cloud.
>
> **Q2:  Adopting part-based feature aggregation module and part-based feature representation to general point cloud models.**
>
> It is an interesting idea to adopt the part-based feature aggregation module and part-based feature representation to general point cloud models. We conduct two experiments for the point cloud classification task on ModelNet40.
>
> 1. We use PointNet++ as the backbone and replace the final max-pooling layer with a part-based feature aggregation module, which improves the classification accuracy from **91.17%** to **91.98%**.
>
> 2. We also adopt part-based feature representation to the general single domain point cloud classification model.  PointNet++ could not provide the point-wise features, so we use PointNet as the backbone and train PDG (PointNet) on ModelNet40. PDG (Point) improves the classification results from **90.19%** to **90.69%** compared with PointNet.
>
> These improvements demonstrate that our proposed part-based feature aggregation module and part-based feature representation are possible to be combined with general point cloud models and further improve their performance for train / test data in the same domain.
>
> **Q3: Related works on dictionary learning in cross-domain problem.**
>
> Thanks for this suggestion. In CDG-UDA [R1], they propose a category dictionary guided unsupervised domain adaptation model for the cross-domain object detection problem. Category-specific dictionaries are learned from the source domain to represent the candidate boxes in the target domain.
>
> In our work, for point cloud domain generalization problem, we find that local geometric structures encoded by part-template features are shared across different domains, which inspires us to align part-level features in different domains to the part-template features. The part-template features served as a dictionary of local geometric structures of 3D shapes.
>
> We will cite these related works in our paper.
>
> [R1] Li, Shuai, et al. Category dictionary guided unsupervised domain adaptation for object detection. In AAAI, 2021.

---

### Meta-Review · Area_Chair_kaDM · 2022-08-27

**Recommendation:** Accept
**Confidence:** Certain

**Metareview:**

The paper works on domain generalization of 3D point cloud classification, and proposes a part-based domain generalization network for the purpose, whose key idea is to build a common feature space of part template and align the part-level features wherein. Three reviewers appreciate the contributions, including the clear motivation, the implicit domain alignment by part-template features, and the proposed part feature aggregation module. They also suggest to improve the paper by clearer definitions of parts, better organization of contrastive learning in the paper, a more complete citation of closely related works, etc.

After discussions between the authors and reviewers, consensus is reached on accepting the paper.  Congratulations!


**Award:**

No

---

### Decision · Program_Chairs · 2022-09-14

Accept